# Characterization of the Key Aroma Constituents in Fry Breads by Means of the Sensomics Concept

**DOI:** 10.3390/foods9081129

**Published:** 2020-08-17

**Authors:** Ola Lasekan, Fatma Dabaj

**Affiliations:** Department of Food Technology, University Putra Malaysia, Serdang UPM 43400, Malaysia; fdabaj@yahoo.co.uk

**Keywords:** aroma constituents, aroma extract dilution analysis, fry bread, odor activity values, solvent-assisted flavor evaporation

## Abstract

The key aroma constituents in the volatile fractions isolated FROM two differently processed fry breads by solvent-assisted flavor evaporation were characterized by an aroma extract dilution analysis (AEDA). Twenty-two compounds were identified with flavor dilution (FD) factor ranges of 2–516. Among them, 13 compounds (FD ≥ 16) were quantified by stable isotope dilution assays and analyzed by odor activity values (OAVs). Of these, 11 compounds had OAVs ≥ 1, and the highest concentrations were determined for δ-decalactone and 2,3-butanedione. Two recombination models of the fry breads showed similarity to the corresponding fry breads. Omission tests confirmed that aroma-active constituents, such as δ-decalactone (oily/peach), 2-acetyl-1-pyrroline (roasty/popcorn-like), 3-methylbutanal (malty), methional (baked potato-like), 2,3-butanedione (buttery), phenyl acetaldehyde (flowery), (*E,E*)-2,4-decadienal (deep-fried), butanoic acid, and 3-methylbutanoic acid, were the key aroma constituents of fry bread. In addition, 3-methoxy-4-vinylphenol (smoky) and 4-hydroxy-2,5-dimethyl-3(2H)-furanone were also identified as important aroma constituents of fry bread.

## 1. Introduction

Fry bread or scone is Native American bread widely consumed in the United States. Fry bread is produced from frozen or unfrozen flat dough which is fried or deep-fried in oil, shortening or lard. Fry breads’ formulation is quite similar to that of bread rolls: flour, sugar, powdered milk, salt, and water. While bread is simply a mixture of flour, water, yeast, and salt with or without butter in the right proportions, kneaded, fermented, and baked in an oven [1]; fry bread formula is devoid of yeast, and the dough is deep-fried at a temperature of 176.7 °C. The quality of fry bread is normally defined by its texture, color and flavor (i.e., the sum of the gustative and olfactory responses observed during food consumption) [2]. Among these qualities, the flavor of fry bread, as in other bread types, is one of the most important factors that determine its acceptance by consumers [3].

The flavor of bread as well as the key aroma compounds responsible for its characteristic flavor has been well documented [4,5,6]. Oftentimes the flavor of bread is brought about by the interaction of a large number of compounds which exhibit different olfactive characteristics. Some of these compounds include pyrazines, aldehydes, esters, ketones, acids, alcohols, hydrocarbons, pyrrolines, furans, etc. [2,3]. Other determinant factors influencing the odor quality of bread are the type of flour, type of fermentation [7], dough improvers [8], and production process [9]. While more than 540 volatiles have been identified in bread [10], only a small fraction of these volatile compounds plays a significant role in the bread’s aroma [1]; interestingly, these small fractions are the ones detectable by the human olfactory receptors. In recent times, the development of extraction methods and analytical procedures has been employed to identify these volatile fractions [4,11,12]. However, most of the methods employed for the quantitation of these volatile fractions are usually carried out with external calibration without necessarily addressing losses during the workup procedure. To address this issue, stable isotope dilution assays (SIDAs) in combination with gas chromatography-mass spectrometry-olfactometry as well as the calculation of odor activity values (OAVs) are now being employed [13,14]. Currently there are many documented reports on the volatile constituents of bread. However, for Native Americans, with an estimated population of 42 million [15], fry bread is primarily a food for special occasion, similar to cake [16]. It can be served as a savory meal topped with cheese or meat. However, the flavor characteristic of this product, which is eaten across the United States, has not been reported to the best of our knowledge.

Therefore, there is a need for a concerted effort to elucidate the flavor compounds in fry bread. This study was therefore aimed at characterizing the key aroma constituents of fry bread using the sensomics approach. The sensomics approach is the best method to date for identifying compounds which play an active role in aroma perception. The approach can also be used to isolate taste components in food along with aroma compounds [17]. In addition, while other flavor analytical techniques rely on separation-based chromatographic methods to quantify the aroma strength of individual compounds in a food matrix, the sensomics approach combines the aforementioned techniques with reconstitution and omission experiments to evaluate the role of specific compounds in the perceived aroma of a mixture. The practical fallout of this approach is the so-called flavor blueprint or flavor signature of a food, i.e., the combinatorial code of the entire set of odor- and taste-active food components in their natural concentrations in food [18]. Furthermore, the sensomics approach has been employed in the characterization of aroma compounds of yeast dough dumpling [14] and the crust of soft pretzels [13].

## 2. Materials and Methods 

### 2.1. Fry Bread Production 

Fry breads were produced by employing the two commonly used methods by Native North Americans. Fry bread can be made from either frozen (−30 °C) or freshly made doughs. The dough recipes contained high protein (13%) (enriched bakers patent flour from Pastry Product, Sdn., Malaysia) (1000 g); warm water (550 g, 50 °C); salt (15 g); granulated sugar (30 g); powdered milk (15 g); and baking powder (30 g). The flour, salt, granulated sugar, and baking powder were mixed separately in a large bowl. The powdered milk was dissolved in the warm water and subsequently added slowly to the dry mixture. The ingredients were introduced into a mixer (Stephan, Hameln, Germany) and mixed for 3 min to produce fluffy dough. The dough mass was divided into 8 golf ball-sized pieces. Four of the golf ball-sized pieces (A) were kept frozen (−30 °C) for 24 h. The other four pieces (i.e., unfrozen) (B) were allowed to rise in a warm spot (29 ± 2 °C) for approximately 20 min. Each of the dough pieces was flattened with a rolling pin to approximately 0.102–0.127 M circular discs and fried in 0.051 to 0.076 M sunflower oil maintained at 180 °C in a mini electric frying pan (Model/SKU 745409792, Helenite). The flattened dough pieces were fried on each side for about 15 s. The golden-brown fry breads (UFBs) (approximately 0.153 M) were removed and drained on a paper towel. The frozen dough pieces (A) were removed after 24 h and kept in a zippered plastic bag inside a fridge overnight to defrost. The thawed dough pieces were flattened into circular discs (0.102–0.127 M) and fried (FBs) as described earlier.

### 2.2. Chemicals 

The pure samples of the following compounds: nonanal, 2,3-butanedione (diacetyl), acetic acid, 3-methylbutanal, butanoic acid, 2-methylpropanoic acid, 3-methylbutanoic acid, pentanoic acid, octanoic acid, and hexanoic acid were purchased from Merck (Darmstadt, Germany). (*E*)-2-nonenal, 2-nonanone, phenyl acetaldehyde, δ-decalactone, 2-methoxy-4-vinylphenol, 4,5-epoxy-(*E*)-2-decanal, (*E,E*)-2,4-decadienal, (*Z*)-2-nonenal, methional, and 4-hydroxy-2,5-dimethyl-3 (2H)-furanone were from Sigma–Aldrich (Taufkirchen, Germany). 1-Octen-3-one was from Symrise (Holzminden, Germany). The following labelled compounds were synthesized according to the literature cited; [^2^H_2_]-Butanoic acid [19]; 3-[^2^H_2_]-methylbutanal [11]; [^2^H_2-5_]-2-acetyl-1-pyrroline [20]; [^13^C_4_]-2,3-butanedione [21]; [^2^H]-(*E,E*)-2,4-decadienal [22]; [^13^C_2_]-acetic acid [22]; [^2^H_2_]-3-methylbutanoic acid [23]; [^2^H_3_]-2-methoxyphenol [24]; [^2^H_5_]-phenyl acetaldehyde [25]; [^13^C_2_]-4-hydroxy-2,5-dimethyl-3(2H)-furanone [26]; [^2^H_2_]-δ-decalactone [27]; [^2^H_2_]-methional [27]. Lastly, [^2^H_3_]-hexanoic acid was from Cambridge Isotope Laboratories (Euriso-top GmbH, Saarbrucken, Germany). 

### 2.3. Isolation of Volatiles from Pulverized Fry Bread for Aroma Extracts Dilution Analysis (AEDA) 

Fry breads (i.e., UFBs and FBs) were each (150 g) sliced into pieces and frozen separately in liquid nitrogen. The frozen bread pieces were pulverized in a warring blender. The volatiles from the pulverized crumb (150 g) were extracted at room temperature (30 ± 2 °C) with dichloromethane (350 mL, 1 h), and the extract was subsequently distilled at 40 °C using the solvent-assisted flavor evaporation (SAFE) distillation protocol [28]. The distillate was treated with aqueous sodium carbonate solution (0.5 mol/L, 50 mL × 3) to remove the acidic volatiles [29]. Furthermore, the aqueous solution was washed with dichloromethane (50 mL), and the organic phases were combined and dried over anhydrous sodium sulphate. It was filtered and concentrated to 1 mL with a small size Vigreux column (40 cm × 1 cm). All analyses were repeated in triplicate.

### 2.4. Analysis of Volatiles

#### 2.4.1. Gas Chromatography-Mass Spectrometry Analysis 

The volatile constituents of the fry bread extracts were analyzed by the GC-MS system (GC-MS, QP-5050A, Shimadzu, Kyoto, Japan) equipped with a (30 m × 0.32 mm, 0.5 μm film thickness) DB-WAX (J & W Scientific, Folsom, USA) column. The extracts (2 µL) were applied by the on-column injection technique at 220 °C. The temperature of the oven was raised from 40 °C. Min^−1^ to 50 °C, held for 2 min isothermally, and then raised from 4 °C. Min^−1^ to 250 °C. The flow rate of the carrier helium was 2.0 mL Min^−1^. The retention indices (RIs) of the compounds were calculated as described previously [29]. 

Mass spectra were recorded in the electron impact positive mode (EI) over a scan range of m/z 40–270 (scan frequency 5.8 Hz) applying an electron energy of 70 eV. The total run time was 45 min. The source and transfer line temperatures were 200 and 240 °C, respectively. Mass spectra were evaluated using the Xcalibur software (Thermos Scientific, Dreieich, Germany). 

#### 2.4.2. Gas Chromatography-Olfactometry (GC-O) 

In order to identify the aroma-active constituents in the fry bread extracts, an olfactory detection port ODP-3 (Gestalt, Mulheim, Germany) which was connected to a Trace Ultra 1300 gas chromatograph (Thermos Scientific, Waltham, MA, USA) was used to conduct the sniffing test. The GC-O system was fitted with a DB-Wax column (30 m × 0.32 mm and 0.5 μm film thickness, J & W Scientific, Folsom, CA, USA). Sniffing was conducted as described previously [30]. Three experienced panelists (two females and a male) with strong gustative and olfactory responses in earlier sessions were used for the sniffing test. The sniffing analysis was divided into three sessions of 20 min, and each assessor participated in the exercise. All analyses were repeated in triplicate by each assessor.

#### 2.4.3. Aroma Extracts Dilution Analysis (AEDA)

The flavor dilution (FD) factors of the aroma constituents were determined by GC-O as reported by Lasekan and Yap [30]. The original extracts (200 µL) containing the neutral/basic volatile constituents obtained from the pulverized fry bread (150 g) were diluted in a stepwise fashion by the addition of dichloromethane as described earlier [30]. Three panelists evaluated all dilutions in triplicate. Only the aroma compounds detected by more than two panelists were recorded.

### 2.5. Aroma Constituents’ Quantification by Stable Isotope Dilution Assays (ACQSIDAs)

Labelled standards (10–60 µg) which were earlier dissolved in dichloromethane (5 mL) were added to each pulverized fry bread sample (100 g). The extract was distilled using the SAFE distillation method described earlier in Section 2.3. Aliquots (0.5 µL) of the concentrates were analyzed by means of two-dimensional GC-MS as described previously [31]. The calibration factor for each compound was determined by analyzing mixtures of the defined quantity of the labelled compounds in five different mass ratios (1:5, 1:3, 1:1, 3:1, and 5:1) using the GC-MS. The obtained response factors from the peak area and the amounts of labelled compound are shown in Table 1.

### 2.6. Aroma Profile Determination

An hour after frying, the pulverized fry breads (i.e., UFBs and FBs) (approximately 8 g with similar crust covering) were placed inside glass beakers (height 7 cm, volume 45 mL) with three random digitals and orthonasally analyzed by panel members at room temperature (29 ± 2 °C). In addition, samples were rotated among panelists to avoid a carry-over effect. The panel was made up of 10 members, aged 24 to 35 years, and composed of seven women and three men. These panelists participated in weekly sensory training sessions for at least a year to be able to recognize and describe different aroma qualities. The sensory analyses were conducted in a sensory room according to international standards (ISO 8589, 2007) [33] with individual booths equipped with uniforms and glare-free white light (D65). Descriptors used were determined in preliminary sensory experiments as described by Steinhause et al. [34]. The reference compounds used as stimuli were: 10 μg L^−1^ of 2-acetyl-1-pyrroline (roasty); 100 μg L^−1^ of 3-methylbutanal (malty); 70 μg L^−1^ of 2,3-butanedione (buttery); 50 μg L^−1^ of 4-vinyl-2-methoxyphenol (smoky); 100 μg L^−1^ of methional (baked potato-like); and 50 μg L^−1^ of δ-decalactone (oily/peach). During evaluation, the panelists had 5 min to rest after each set of samples was tested. All samples were repeated in triplicate. The intensities of the attributes were rated on a 7 point linear scale from 0 (not perceivable) to 3 (strongly perceivable) in steps of 0.5 by the panelists. The sensory data were analyzed in triplicate using the Student’s *t*-test, and statistical analyses were performed using SPSS 20.0 (SPSS Inc., Chicago, IL, USA). In addition, all procedures performed in studies involving human participants were in accordance with the ethical standards of the institutional and/or national research committee and with the 1964 Helsinki Declaration and its later amendments or comparable ethical standards. The study protocol and consent procedure received ethical approval from the Institutional Review Board (IRB) of the University Putra Malaysia. Informed consent was obtained from all individual participants included in the study. 

### 2.7. Aroma Model Recombinants and Omission Tests of the Fry Breads (UFBs and FBs)

Reference standards of aroma constituents with OAVs above 1 (Table 2) were prepared in ethanolic solution [35]. The combined ethanolic stock solutions of the 11 aroma compounds (500 μL) with OAVs ≥ 1 was mixed with 30 mL of citrate buffer (pH 5.6; 0.1 mol L^−1^) and 30 g of free corn starch in a closed Teflon cup. The Teflon cup was stirred continuously for 15 min at room temperature (i.e., 29 °C). A triangle test was performed to determine the significance of one odorant on the aroma recombination models (UFBs and FBs) reported in (Table 4). For each of the models, a glass of the mixture (20 mL) was prepared by omitting one or a group of selected odorants from the complete recombination model (Table 5). This mixture and two other glasses containing the complete recombination models were presented to the sensory panel in a triangle test [36]. The results of the triangle tests were analyzed by comparing the total number of correct responses with the minimum number of responses required for statistical significance (ISO 4120, 2004) [37]. Panel performance was obtained by applying analysis of variance (ANOVA) to the sensory profile data. The data were analyzed using SAS statistical software (SAS Institute, Inc., Cary, NC, USA, 1996). The significance α was calculated according to the method of Callejo et al. [36]. In addition, each of the aroma models was evaluated orthonasally in comparison with the corresponding fry bread (i.e., UFB or FB) as described above (Section 2.6.).

## 3. Results and Discussion

### 3.1. The Aroma-Active Constituents of Fry Breads

A total of 22 aroma-active constituents were identified in the fry breads (i.e., fry breads produced from frozen dough, (FB) and those made from unfrozen dough (UFB)). Among these compounds, seven acids, eight aldehydes, four ketones, one heterocyclic compound, one furan, and a phenol were positively identified (Table 2). These aroma constituents exhibited an array of odor nuances such as malty, buttery, roast-like, baked potato-like, sweaty, deep-fried, caramel, smoky, and oily/peach-like. To establish differences among the flavors of the fry bread samples, the volatile fractions of the fry bread extracts were subjected to AEDA. The results of the AEDA and the identification experiments carried out are shown in Table 2. In the neutral–basic fraction (Table 2), the following aroma-active constituents, δ-decalactone (FD = 256), 2-methoxy-4-vinyl phenol (FD = 64), methional (FD = 64), 3-methylbutanal (FD = 64), 2,3-butanedione (FD = 32), and (*E,E*)-2,4-decadienal (FD = 32), produced the highest FD factors in the UFB. This group was followed by 2-acetyl-1-pyrroline (FD = 16) and phenyl acetaldehyde (FD = 16). For the acid fraction, acetic acid (FD = 512), butanoic acid (FD = 64), 3-methylbutanoic acid (FD = 64), 4-hydroxy-2,5- dimethyl-3(2H)-furanone (FD = 32), and hexanoic acid (FD = 16) recorded the highest FD factors. In the case of the fry breads produced from the frozen dough (FB), the highest FD values were recorded for the following compounds detected in the neutral–basic fraction: δ-decalactone (FD = 256), 2-methoxy-4-vinyl phenol (FD = 64), 3-methylbutanal (FD = 64), methional (FD = 64), 2,3-butanedione (FD = 32), and (*E,E*)-2,4-decadienal (FD = 32). On the other hand, acetic acid, butanoic acid, 3-methylbutanoic acid, and 4-hydroxy-2,5-dimethyl-3(2H) furanone recorded the highest FD values (32–256) in the acid fraction of the fry breads (FB).

### 3.2. Aroma Quantitation in the Fry Breads

To further evaluate the contribution of each aroma compound identified with high FD factors (i.e., FD factors ≥ 16) to the overall aroma of the fry bread, the compounds were subjected to quantitation using SIDA. The quantitation results revealed that δ-decalactone with the oily/peach note presented significantly (*p* < 0.05) high concentrations, with 1916 μg kg^−1^ and 1908 μg kg^−1^ in the UFB and FB, respectively (Table 3). It was followed by 2,3-butanedione (924–925 μg kg^−1^), acetic acid (668–716 μg kg^−1^), 3-methylbutanoic acid (618–621 μg kg^−1^), and butanoic acid (348–350 μg kg^−1^) in FB and UFB, respectively. Slightly lower concentrations were obtained for 4-hydroxy-2,5-dimethyl-3(2H)-furanone, 3-methylbutanal, (*E,E*)-2,4-decadienal, 2-methoxy-4-vinylphenol, phenyl acetaldehyde, and methional. 2-Acetyl-1-pyrroline recorded a value < 3.0 μg.kg^−1^. It is worthy of note that δ-decalactone, which had the highest concentration in both UFB and FB, had also been identified as a key aroma constituent in most fat-containing foods [38] such as butter oil [39] and puff pastries [40]. 

In addition, nearly all of the aroma-active constituents, disregarding their boiling point, polarity or functional group, decreased slightly (i.e., <7%) in the fry breads produced from the frozen doughs (FB) (Table 3). The only exception to this were 2,3-butanedione, 2-acetyl-1-pyrroline, and 2-methoxy-4-vinylphenol. The slight decreases (i.e., <7%) obtained between the FB and UFB were probably due to the frozen dough used in the production of FB. Studies have shown that freezing often results in the disruption of the gluten structure as a result of water migration brought about by crystal formation. This phenomenon results in the dissociation of starch granules from the gluten network [41]. The disruption of the starch granules is known to influence the interaction of starch with the volatile compounds as well as the retention of volatile compounds [42] Overall, the aldehydes and acids constituted the largest number of aroma constituents (FD ≥ 16) (Table 3) in the fry breads. Aldehydes are typical lipid oxidation products that are associated with characteristic aroma of whole wheat bread [43]. In addition, aldehydes, such as 3-methylbutanal (malty) and phenyl acetaldehyde, identified in the fry breads can also be formed during the Maillard reaction [44]. Another route for the formation of the aldehydes could be through the amino acid biosynthetic pathway in which the aldehydes formed during the Ehrlich pathway is oxidized to their corresponding acids such as 3-methylbutanoic acid by aldehyde dehydrogenase [45]. It should be mentioned that acetic acid is also a well-known product of the Maillard reaction.

### 3.3. The Aroma-Active Constituents (FD ≥ 16) and Their Potencies

To elucidate the contribution and potency of each compound to the overall aroma of the fry breads, the OAVs of the aroma constituents with FD ≥ 16 were calculated on the basis of their odor thresholds in starch (Table 3). The fry breads exhibited more potencies for the following aroma notes: roasty/popcorn-like, baked-potato-like, buttery, and sweaty as revealed by the high values obtained for the OAVs of their corresponding compounds (e.g., 2-acetyl-1-pyrroline, (329–343); methional, (267–278); 2,3-butanedione, (142); and 3-methylbutanoic acid (103–104) for FB and UFB respectively) (Table 3). For instance, 2-acetyl-1-pyrroline exceeded its threshold by a factor of 343 in UFB and by 329 in FB (Table 3). Methional exceeded its threshold by a factor of 278 in UFB and by 267 in FB. A similar trend was obtained for 2,3-butanedione, 3-methylbutanoic acid, (*E,E*)-2,4-decadienal, and 4-hydroxy 2,5-dimethyl-3(2H)-furanone. Lower potencies were recorded for 3-methylbutanal (malty), 2-methoxy-4-vinylphenol (smoky), and butanoic acid. However, acetic acid and hexanoic acid had OAVs below 1, and it is assumed that these compounds did not contribute to the overall aroma of the fry breads.

### 3.4. Aroma Profile Analysis and Aroma Simulation Models of the Fry Breads (UFBs and FBs)

The results of the sensory evaluation of the different fry breads (UFBs and FBs) are shown in (Figure 1A, Table 4). The aroma profiles of the fry breads were characterized as roasty/popcorn-like, malty, buttery, baked potato-like, smoky, sweaty, deep-fried, and oily/peach-like. However, with the exception of the malty and buttery notes, the aroma profiles of the UFBs and FBs were quite similar. The malty and buttery nuances in the FB samples were less intense as compared to that of the UFB samples (Table 4). The Duncan’s multiple comparison test results (Table 4) revealed that the eight attributes (roasty/popcorn-like, malty, buttery, baked potato-like, smoky, sweaty, deep-fried, and oily/peach-like) with different superscripts seemed to well explain their aroma characteristics. To validate this observation, recombination experiments were performed by mixing solutions of the pure reference compounds in the same amounts as obtained for both UFB and FB (Table 5). The recombination models were evaluated in parallel with the UFB and FB samples. Sensory results revealed that the recombinant models imitated well the flavor of the fry breads (Figure 1B,C). In addition, the roasty/popcorn aroma note was perceived as equally intense in the aroma models as well as in the fry breads.

### 3.5. Omission Tests

To assess the contribution of individual compound to the overall aroma of the fry bread (UFB and FB), omission tests were conducted on the fry bread aroma models (Table 6) [47]. In this study, 7 aroma omission models (M1–M7) comprised of either a single or group of compounds were prepared. Each of the omission models was evaluated in triangular experiments with two complete recombination models (Table 6). The results revealed that, when the entire group of aldehydes (M1) was omitted, their omission from the complete recombination model could be detected by 9 out of the 10 assessors. This is an indication of the importance of these aldehydes (i.e., 3-methylbutanal, methional, phenyl acetaldehyde and (*E,E*)-2,4-decadienal) in the overall aroma of the fry breads.

In the second group, the acids (butanoic acid and 3-methylbutanoic acid) were omitted. The result of the omission of all the acids from the complete recombination models showed that 8 out of the 10 assessors were able to detect between the omission model and the complete recombination models. This shows that the acids also influence the overall aroma of the fry breads. Similar results were obtained when all ketones (M3) were omitted. Omission of single compounds, such as 2-methoxy-4-vinylphenol (smoky) (M4) or 4-hydroxy-2,5-dimethyl-3(2H)-furanone (caramel-like) (M6), resulted in less significant (*p* ≤ 0.05) reductions in the characteristics aroma of the fry breads. Only 7 out of the 10 assessors could detect the omission of either compound. However, the omission of 2-acetyl-1-pyrroline (popcorn-like) (M5) or δ-decalactone (oily/peach) (M7) resulted in a highly significant (*p* ≤ 0.001) reduction in the characteristic aroma of the fry breads. In the case of 2-acetyl-1-pyrroline, all 10 assessors were able to detect its omission from the complete recombination models.

## 4. Conclusions

In conclusion, differences in the aroma-active constituents of fry breads produced from frozen doughs (FBs) and freshly made doughs (UFBs) were characterized for the first time. A total of twenty-two aroma constituents were identified in the fry breads. The results of the OAVs and sensory studies showed that the aroma profiles of the fry breads were characterized as roasty/popcorn-like, malty, buttery, baked potato-like, smoky, sweaty, deep-fried, and oily/peach-like. However, with the exception of the malty and buttery notes, the aroma profiles of UFBs and FBs were quite similar, and the malty and buttery nuances in the FB samples were less intense as compared to that of the UFB samples. Aroma-active constituents, such as δ-decalactone (oily/peach), 2-acetyl-1-pyrroline (roasty/popcorn-like), 3-methylbutanal (malty), methional (baked potato-like), 2,3-butanedione (buttery), phenyl acetaldehyde (flowery), (*E,E*)-2,4-decadienal (deep-fried), butanoic acid, and 3-methylbutanoic acid, were identified as the key aroma constituents of fry bread. In addition, 3-methoxy-4-vinylphenol (smoky) and 4-hydroxy-2,5-dimethyl-3(2H)-furanone (caramel-like) were identified as important aroma constituents of fry bread. Finally, these findings establish a basis for further work on the identification of the taste-active food components in fry breads as well as consumers’ preferences for the differently produced fry breads.

## Figures and Tables

**Figure 1 foods-09-01129-f001:**
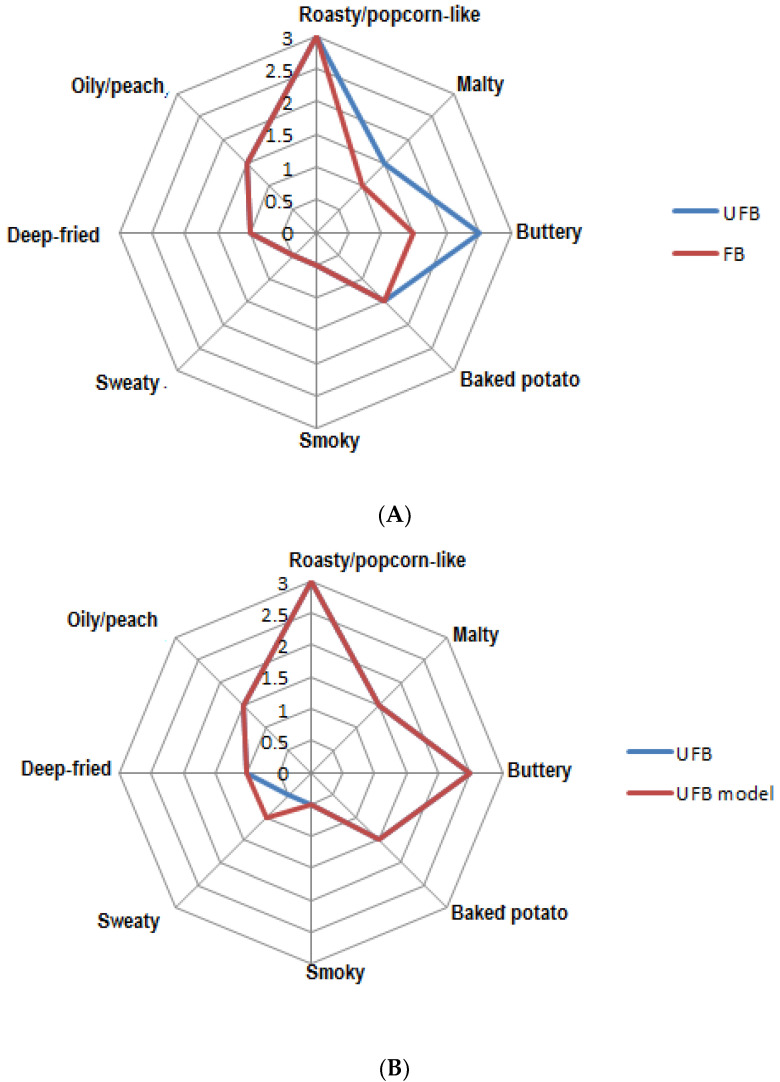
(**A**): Aroma profiles of fry breads produced from unfrozen dough (UFB) (blue line) and frozen dough (FB) (orange line). (**B**): A comparative aroma profile of FB (blue line) and its aroma model (FB-m) (orange line). (**C**): Aroma profile of UFB (blue line) and its aroma model (UFB-m) (orange line).

**Table 1 foods-09-01129-t001:** Selected ions and calibration factors used for the quantification of aroma compounds in fry bread by stable isotope dilution assays.

Number	Compounds ^a^	Selected Ions (m/Z)	Internal Standards	Selected Ions (m/z)	Calibration Factor
1	Acetic acid	61	[^13^C_2_]-acetic acid	63	1.00
2	3-Methylbutanoic acid	60	[^2^H_2_]-3-methylbutanoic acid	62	1.00
3	2,3-Butanedione	87	[^2^H_2_]-2,3-butanedione	91	0.90
4	3-Methylbutanal	87	[^2^H_2_]-3-methylbutanal	89	1.00
5	Butanoic acid	89	[^2^H_2_]-butanoic acid	91	0.95
6	Methional	105	[^2^H_2_]-methional	108	1.00
7	2-Acetyl-1-pyrroline	112	[^2^H_2_]-2-acetyl-1-pyrroline	114	1.00
8	Hexanoic acid	117	[^2^H_2_]-hexanoic acid	120	0.95
9	Phenyl acetaldehyde	121	[^2^H_2_]-2-phenyl acetaldehyde	123	0.85
10	4-Hydroxy-2,5- dimethyl-3(2H)- furanone	129	[^13^C_2_]-4-hydroxy-2,5- dimethyl-3(2H)-furanone	131	1.00
11	2-Methoxy-4-vinylphenol	150	[^13^C_6_]-2-methoxy-4-vinylphenol	156	0.85
12	*(E,E*)-2,4-decadienal	153	[^2^H_2_]-(E,E)-2,4-decadienal	156	0.97
13	δ-Decalactone	171	[^2^H_2_]-δ-decalactone	173	1.00

^a^ The compounds and calibration factors were determined as reported previously by Lasekan, Buettner, and Christlbauer (2007) [31] and Guth and Grosch (1993) [32], respectively.

**Table 2 foods-09-01129-t002:** Aroma-active constituents in two differently produced fry breads.

No	Compound ^a^	Odor Note	Retention Index on DB-Wax	Fraction	Flavor Dilution * (UFB)	Flavor Dilution * (FB)
1	3-Methylbutanal	Malty	900	NB	64	64
2	2,3-Butanedione (diacetyl)	Buttery	976	NB	32	32
3	1-Octen-3-one	Mushroom	1316	NB	4	4
4	2-Acetyl-1-pyrroline	Roasty	1325	NB	16	16
5	2-Nonanone	Soapy/fruity	1388	NB	8	8
6	Nonanal	Fatty	1391	NB	8	8
7	Acetic acid	Vinegar	1453	A	512	256
8	Methional	Baked potato	1478	NB	64	64
9	(*Z*)-2-Nonenal	Fatty/green	1511	NB	2	2
10	2-Methypropanoic acid	Sweaty	1514	A	8	8
11	(*E*)-2-Nonenal	Cucumber	1542	NB	4	4
12	Butanoic acid	Sweaty	1638	A	64	32
13	Phenyl acetaldehyde	Flowery	1650	NB	16	16
14	3-Methylbutanoic acid	Sweaty	1674	A	64	64
15	Pentanoic acid	Sweaty	1698	A	8	8
16	(*E,E*)-2,4-Decadienal	Deep-fried	1710	NB	32	32
17	Hexanoic acid	Sweaty	1795	A	16	16
18	4,5-Epoxy-(E)-2-decanal	Metallic	2010	NB	4	4
19	4-Hydroxy-2,5-dimethyl3(2H)-furanone	Caramel	2030	A	32	32
20	Octanoic acid	Soapy/fatty	2064	A	8	8
21	δ-Decalactone	Oily/peach	2112	NB	256	256
22	2-Methoxy-4-vinylphenol	Smoky	2203	NB	64	64

^a^ The compounds were identified by comparing the mass spectra through the mass spectra/electron ionization (MS/EI), the retention indices was detected on bonded low bleed wax capillary column (DB-Wax), and the odor note perceived at the sniffing port. * Flavor dilution (FD) determined by aroma extract dilution analysis (AEDA) for fry breads produced from unfrozen (UFB) and frozen (FB) doughs. NB: Neutral–basic fraction, A: acidic fraction.

**Table 3 foods-09-01129-t003:** Concentration, odor thresholds, and odor activity values (OAVs) of aroma-active constituents (FD ≥ 16) in fry breads.

No	Compound	Concentration (μg kg^−1^ wet wt.)		Threshold in Starch * (μg kg^−1^)	(OAVs)	
		UFB	FB		UFB	FB
1	3-Methylbutanal	240 ± 9.0 ^a^	228 ± 12.0 ^b^	32	7.5	7.0
2	2,3-Butanedione	925 ± 20.0 ^a^	924 ± 15.0 ^a^	6.5	142	142
3	2-Acetyl-1-pyrroline	2.5 ± 0.1 ^a^	2.4 ± 0.1 ^a^	0.0073	343	329
4	Acetic acid	716 ± 16.5 ^a^	668 ± 12.0 ^b^	31,140	<1	<1
5	Methional	75 ± 9.2 ^a^	72 ± 11.5 ^b^	0.27	278	267
6	Butanoic acid	350 ± 8.8 ^a^	348 ± 5.5 ^a^	100	3.5	3.5
7	Phenyl acetaldehyde	107 ± 5.0 ^a^	102 ± 7.1 ^b^	ND	ND	ND
8	3-Methylbutanoic acid	621 ± 13.0 ^a^	618 ± 10.0 ^b^	6	104	103
9	(*E,E*)-2,4-Decadienal	147 ± 7.6 ^a^	144 ± 4.6 ^b^	2.7	54	53
10	Hexanoic acid	265 ± 9.0 ^a^	259 ± 6.5 ^b^	11,000	<1	<1
11	4-Hydroxy-2,5-dimethyl3(2H)-furanone	265 ± 8.0 ^a^	263 ± 7.1 ^b^	13	20	20
12	δ-Decalactone	1916 ± 31.0 ^a^	1908 ± 23.0 ^b^	ND	ND	ND
13	2-Methoxy-4-vinylphenol	113 ± 5.0 ^a^	113 ± 3.0 ^a^	18.3	6	6

* Thresholds in starch (Rychlik, Schieberle, and Grosch (1998) [46], OAV odor activity values on the basis of threshold in starch. UFB, fry breads produced from unfrozen doughs; FB, fry breads produced from frozen doughs; ND not detectable. Different letters within the same row represent significant differences.

**Table 4 foods-09-01129-t004:** The mean scores of the eight attributes of the fry breads and the aroma models generated.

Sensory Attributes	Fry Breads			Mean Scores of Fry Breads and Their Models			
	UFB	FB	*p*-Value	UFB	UFB Model	FB	FB Model
Roasty/Popcorn	3.0 ± 0.32	3.0 ± 0.51	0.01	3.0 ± 0.4 ^a^	3.0 ± 0.3 ^a^	3.0 ± 0.2 ^A^	3.0 ± 0.4 ^A^
Malty	1.5 ± 0.06	1.0 ± 0.10	0.00	1.5 ± 0.1 ^a^	1.5 ± 0.4 ^a^	1.0 ± 0.0 ^A^	1.0 ± 0.1 ^A^
Buttery	2.5 ± 0.40	1.5 ± 0.03	0.07	2.5 ± 0.4 ^a^	2.5 ± 0.8 ^a^	1.5 ± 0.1 ^A^	1.5 ± 0.3 ^A^
Baked potato	1.5 ± 0.17	1.5 ± 0.10	0.00	1.5 ± 0.2 ^a^	1.5 ± 0.1 ^a^	1.5 ± 0.1 ^A^	1.5 ± 0.3 ^A^
Smoky	0.5 ± 0.00	0.5 ± 0.00	0.00	0.5 ± 0.0 ^a^	0.5 ± 0.1 ^a^	0.5 ± 0.0 ^A^	0.5 ± 0.1 ^A^
Sweaty	0.5 ± 0.01	0.5 ± 0.00	0.00	0.5 ± 0.0 ^b^	1.0 ± 0.3 ^a^	0.5 ± 0.1 ^A^	0.5 ± 0.0 ^A^
Deep-fried	1.0 ± 0.02	1.0 ± 0.00	0.00	1.0 ± 0.1 ^a^	1.0 ± 0.1 ^a^	1.0 ± 0.1 ^A^	1.0 ± 0.0 ^A^
Oily/peach	1.5 ± 0.12	1.5 ± 0.04	0.00	1.5 ± 0.2 ^a^	1.5 ± 0.3 ^a^	1.5 ± 0.1 ^A^	1.5 ± 0.4 ^A^

^A, B, C: a, b, c,^ Different letters within the same row represent significant differences (*p* < 0.05) (n = 30, 10 panelists with three replications), *p*-values among fry bread samples obtained by Student’s *t*-test. UFB, fry bread produced from unfrozen dough; FB, fry bread from frozen dough.

**Table 5 foods-09-01129-t005:** Aroma model compositions for fry breads (UFB and FB).

No	Compounds	Odor Notes	Concentration (μg L^−1^) *	
			UFB	FB
1	3-Methylbutanal	Malty	240	228
2	2,3-Butanedione	Buttery	925	924
3	2-Acetyl-1-pyrroline	Popcorn/roast	2.5	2.4
4	Methional	Baked potato	75	72
5	Butanoic acid	Sweaty	350	348
6	Phenyl acetaldehyde	Flowery/honey	107	102
7	3-Methylbutanoic acid	Sweaty	621	618
8	(*E,E*)-2,4-Decadienal	Deep-fried	147	144
9	4-Hydroxy-2,5-dimethyl-3(2H)- furanone	Caramel-like	265	263
10	2-Methoxy-4-vinylphenol	Smoky	113	113
11	δ-Decalactone ^a^	Oily/peach	1916	1908

* Ethanolic solutions of aroma compounds dissolved in free corn starch. ^a^ δ-Decalactone was added because of its high concentrations.

**Table 6 foods-09-01129-t006:** Omission analysis on the fry bread aroma models (UFB and FB).

Odorant Groups	Aroma Notes	Compounds Omitted	No of Correct Judgments ^a^ UFB FB		Significance ^b^
Aldehydes (M1)	Malty, baked potato-like, flowery, deep-fried	3-Methylbutanal, Methional, Phenyl acetaldehyde, (*E,E*)-2,4-Decadienal,	9/10	9/10	***
Acids (M2)	Sweaty	Butanoic acid, 3-Methylbutanoic acid	8/10	8/10	**
Ketones (M3)	Buttery, Caramel-like	2,3-Butanedione, 4-Hydroxy-2,5-dimethyl-3(2H)-furanone	8/10	8/10	**
Phenols (M4)	Smoky	2-Methoxy-4-vinylphenol	7/10	7/10	*
(M5)	Popcorn/roast	2-Acetyl-pyrroline	10/10	10/10	***
M6	Caramel-like	4-Hydroxy-2,5-dimethyl-3(2H)-furanone	7/10	7/10	*
M7	Oily/peach	δ-Decalactone	8/10	8/10	**

^a^ Number of correct judgments from 10 assessors; ^b^ Significance: * significant (α ≤ 0.05); ** highly significant (α ≤ 0.01); *** very highly significant (α ≤ 0.001); M1–M7 Models. UFB, fry bread from unfrozen dough; FB, fry bread from frozen dough.

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
