# Peer review of "Characterization of the Key Aroma Constituents in Fry Breads by Means of the Sensomics Concept"

_foods, 2020, doi:10.3390/foods9081129_

Round 1
Reviewer 1 Report
Dear authors,
The manuscript is clear and easy to follow. The results shown support your concluding comments and are interesting for readers to know.
I have very few suggestions regarding the manuscript, observations you may consider in order to improve the quality of your work:
Lines 39 and 40: why that difference in names “salt and sodium chloride”?
Line 67: play
Line 189: no reference aroma for g-decalactone? I would add it as it is done for the other compounds
Do you have any results on consumer preferences? Meaning if any of the samples were preferred over the other? Differences in aroma composition in frozen fry breads would eventually mean less quality? That would sum up your whole experimental research with an additional market-perspective opinion.
Author Response
Reviewer 1
Dear authors,
The manuscript is clear and easy to follow. The results shown support your concluding comments and are interesting for readers to know.
I have very few suggestions regarding the manuscript, observations you may consider in order to improve the quality of your work:
Lines 39 and 40: why that difference in names “salt and sodium chloride”?
This observation is noted and we have opted to use ‘salt’ instead of sodium chloride (see line 40)
Line 67: play
This has been corrected (Line 70)
Line 189: no reference aroma for g-decalactone? I would add it as it is done for the other compounds
The aroma reference oily/peach has been added
Do you have any results on consumer preferences? Meaning if any of the samples were preferred over the other? Differences in aroma composition in frozen fry breads would eventually mean less quality? That would sum up your whole experimental research with an additional market-perspective opinion.
We don’t have analysis on consumer preference now. However, consumer preference is being addressed in our next study on taste-active food components of fry bread.

Reviewer 2 Report
Keywords. It is not suggested to use acronyms in keywords, please use the long form of the expressions.
L43-44: bracket closing is missing
L42-43. The authors define texture, color and flavor as the most important quality factors. In the next sentence, however, they state that aroma is one of the most important among them. Please clarify.
L52. Improvers?
L63-64. The authors talk about breads and I miss the connection, why is it important to analyze fry breads. Use some consumption data, prove the lack of scientific papers, etc. to show that analysis of fry breads is a new topic.
I miss the detailed introduction of sensomics. Since it is part of the title, it would be beneficial to introduce the concept and to list the most relevant papers using sensomics.
L82. Either – either
L83-84. Please clarify the ingredients.
L114, L125, L139, L216 Sentence is not closed
L116. Define AEDA here
L121. Define SAFE here
L185. Was the light D65? If yes, please add. Did the sensory lab meet the relevant ISO standards? If yes, please add. Were the assessors trained according to the relevant ISO standard? If yes, please add.
L191. How were the samples coded (three-digit random numbers are the standard method)? Were the samples rotated among panelists to avoid carry-over effect?
L204. Should be numbered as Table 2
Table 2: last columns should be omitted since all rows list the same information
From Table 4 it seems that the authors compared 2 samples based on multiple sensory attributes. If it is true, than the use of ANOVA should be avoided and t-tests should have been used. Furthermore, sensory data usually does not follow normal distribution, therefore nonparametric data analysis should have been used. Please, modify.
Figure 1. the figure is useful but the presentation should be improved. The Figure 1B and C looks like if FB and UFB are missing (probably overlapping).
L354: bracket opening is missing
L357: it looks like the authors used triangle test. Please add this to the methods section. It would be important to add which evaluation method did the authors used for the triangular test.
Conclusion section is not a conclusion but a summary of the work. Conclusion section should address the obtained results and should introduce how these results could be used for future studies and what should be the further steps. Please rewrite conclusions.
The paper deals with an interesting topic and uses advanced analytical and sensory methods. However, the many methods are sometimes hard to follow. I suggest adding a workflow which introduces the steps the authors followed. Additionally, the text should be improved to support this golden line.
Author Response
Reviewer 2
Keywords. It is not suggested to use acronyms in keywords, please use the long form of the expressions.
Your suggestion is noted and the acronyms have been written in full.
L43-44: bracket closing is missing
Correction effected (line 44)
L42-43. The authors define texture, color and flavor as the most important quality factors. In the next sentence, however, they state that aroma is one of the most important among them. Please clarify.
The term flavour is a combination of taste and aroma. Therefore aroma is a subset of flavour. We have made necessary correction in the text
L52. Improvers?
We ought to have included dough. This has been corrected (line 52)
L63-64. The authors talk about breads and I miss the connection, why is it important to analyze fry breads. Use some consumption data, prove the lack of scientific papers, etc. to show that analysis of fry breads is a new topic.
A large population of Native American consume fry bread across the United States and there are scanty scientific papers on this product (see lines 62-66)
I miss the detailed introduction of sensomics. Since it is part of the title, it would be beneficial to introduce the concept and to list the most relevant papers using sensomics.
Relevant papers have been added such as: the characterization of aroma compounds of yeast dough dumpling [14] and the crust of soft pretzels [13].
L82. Either – either
The first either has been deleted line 84
L114, L125, L139, L216 Sentence is not closed
DONE
L116. Define AEDA here
DONE
L121. Define SAFE here
DONE
L185. Was the light D65? If yes, please add. Did the sensory lab meet the relevant ISO standards? If yes, please add. Were the assessors trained according to the relevant ISO standard? If yes, please add.
D65 was used and this has been stated in the text (line 187). The sensory lab also meet the required standard (ISO 8589, 2007) (line 186)
L191. How were the samples coded (three-digit random numbers are the standard method)? Were the samples rotated among panelists to avoid carry-over effect?
3-digit random numbers were used and samples were also rotated (lines 181-182)
L204. Should be numbered as Table 2
Corrected effected
Table 2: last columns should be omitted since all rows list the same information
DONE
From Table 4 it seems that the authors compared 2 samples based on multiple sensory attributes. If it is true, than the use of ANOVA should be avoided and t-tests should have been used. Furthermore, sensory data usually does not follow normal distribution, therefore nonparametric data analysis should have been used. Please, modify.
Suggestion is well noted and the data have been subjected to Student’s T-test (Table 4)
Figure 1. the figure is useful but the presentation should be improved. The Figure 1B and C looks like if FB and UFB are missing (probably overlapping).
The Figures have been re-drawn. FB and UFB are overlapping in Fig 1B and C.
L354: bracket opening is missing
The bracket opening has been added
L357: it looks like the authors used triangle test. Please add this to the methods section. It would be important to add which evaluation method did the authors used for the triangular test.
The results of the Triangle tests were analysed by comparing the total number of correct responses with the minimum number of responses required for statistical significance (ISO, 4120, 2004). Panel performance was obtained by applying analysis of variance (ANOVA) to the sensory profile data. The data were analysed using SAS Statistical software (SAS Institute, Inc. 1996). The significance α was calculated according to the method of Callejo et al, [34] (lines 213-218)
Conclusion section is not a conclusion but a summary of the work. Conclusion section should address the obtained results and should introduce how these results could be used for future studies and what should be the further steps. Please rewrite conclusions.
DONE. Further work on the identification of the taste-active food components in the fry breads as well as consumers’ preferences for the differently produced fry breads would need to be studied
The paper deals with an interesting topic and uses advanced analytical and sensory methods. However, the many methods are sometimes hard to follow. I suggest adding a workflow which introduces the steps the authors followed. Additionally, the text should be improved to support this golden line.
A workflow sheet has been added

Round 2
Reviewer 2 Report
Dear Authors,
As I read your answers it is clear that you addressed all my issues and corrected them carefully. However, the submitted version of the paper contains only comments, and I could not find the new figures you mentioned in your answers.
I gave a minor just because to see the corrected version, which has all your corrections and the new version.
Author Response
Reviewer 2
Dear Authors,
As I read your answers it is clear that you addressed all my issues and corrected them carefully. However, the submitted version of the paper contains only comments, and I could not find the new figures you mentioned in your answers.
The new figures 1A, B & C have been re-drawn. FB and UFB are overlapping in Fig 1B and C. (lines 331-354)
I gave a minor just because to see the corrected version, which has all your corrections and the new version.
According to the editor Ms Lynn Liu, any revisions should be
clearly highlighted, for example using the "Track Changes" function in
Microsoft Word, so that they are easily visible to the editors and reviewers. That is why all corrections were done using the “Track Changes’. Do you want us to remove them again?
